

# Discrimination of *Picea chihuahuana* Martinez populations on the basis of climatic, edaphic, dendrometric, genetic and population traits

Iliana Karina Dominguez-Guerrero[1], Samantha del Rocío Mariscal-Lucero[1], José Ciro Hernández-Díaz[1], Berthold Heinze[2], José Ángel Prieto-Ruiz[3] and Christian Wehenkel[1]

[1] Instituto de Silvicultura e Industria de la Madera, Universidad Juárez del Estado de Durango, Durango, Mexico
[2] Federal Research Centre for Forests, Natural Hazards and Landscape (BFW), Vienna, Austria
[3] Facultad de Ciencias Forestales, Universidad Juárez del Estado de Durango, Durango, Mexico

Corresponding author
Christian Wehenkel,
wehenkel@ujed.mx

## ABSTRACT

**Background.** *Picea chihuahuana*, which is endemic to Mexico, is currently listed as "Endangered" on the Red List. Chihuahua spruce is only found in the Sierra Madre Occidental (SMO), Mexico. About 42,600 individuals are distributed in forty populations. These populations are fragmented and can be classified into three geographically distinct clusters in the SMO. The total area covered by *P. chihuahuana* populations is less than 300 ha. A recent study suggested assisted migration as an alternative to the *ex situ* conservation of *P. chihuahuana*, taking into consideration the genetic structure and diversity of the populations and the predictions regarding the future climate of the habitat. However, detailed background information is required to enable development of plans for protecting and conserving species and for successful assisted migration. Thus, it is important to identify differences between populations in relation to environmental conditions. The genetic diversity of populations, which affect vigor, evolution and adaptability of the species, must also be considered. In this study, we examined 14 populations of *P. chihuahuana*, with the overall aim of discriminating the populations and form clusters of this species.

**Methods.** Each population was represented by one $50 \times 50$ m plot established in the center of its respective location. Climate, soil, dasometric, density variables and genetic and species diversities were assessed in these plots for further analyses. The putatively neutral and adaptive AFLP markers were used to calculate genetic diversity. Affinity Propagation (AP) clustering technique and $k$-means clustering algorithm were used to classify the populations in the optimal number of clusters. Later stepwise binomial logistic regression was applied to test for significant differences in variables of the southern and northern *P. chihuahuana* populations. Spearman's correlation test was used to analyze the relationships among all variables studied.

**Results.** The binomial logistic regression analysis revealed that seven climate variables, the geographical longitude and sand proportion in the soil separated the southern from northern populations. The northern populations grow in more arid and continental conditions and on soils with lower sand proportion. The mean genetic diversity using all AFLP studied of *P. chihuahuana* was significantly correlated with the mean temperature in the warmest month, where warmer temperatures are associated to larger genetic

diversity. Genetic diversity of *P. chihuahuana* calculated with putatively adaptive AFLP was not statistically significantly correlated with any environmental factor.

**Discussion**. Future reforestation programs should take into account that at least two different groups (the northern and southern cluster) of *P. chihuahuana* exist, as local adaptation takes place because of different environmental conditions.

## INTRODUCTION

*Picea chihuahuana* Mtz. (Chihuahua spruce), which is endemic to Mexico, is currently listed as ''Endangered'' on the Red List of the International Union for the Conservation of Nature and Natural Resources (*IUCN, 2013*) and in the official Mexican normativity on endangered species (NOM-059-SEMARNAT-2010) (*SEMARNAT, 2010*). This tree species grows at elevations between 2,150 and 2,990 m in areas with mean annual temperatures of $9-12$ °C and precipitation ranging from around 600 to 1,300 mm (provenance's climate inferred from *Sáenz-Romero et al., 2010*).

Chihuahua spruce is only found in the Sierra Madre Occidental (SMO), Mexico. About 42,600 individuals are distributed in forty populations (*Farjon, Page & Schellevis, 1993*; *Ledig et al., 2000*; *Wehenkel & Sáenz-Romero, 2012*). The species is specifically located in the states of Chihuahua (in the municipalities of Bocoyna, Temosachi, Guerrero, and Balleza) and Durango (in the municipalities of El Mezquital, Pueblo Nuevo, San Dimas, Canelas, and Guanacevi) (*Ledig et al., 2000*). The populations are fragmented and can be classified into three geographical clusters in the SMO of the two States (south, center, and north), each group separated by a distance of about 300 km (*Mendoza-Maya et al., 2015*). The total accumulated area covered by Chihuahua spruce populations is less than 300 ha (*Simental-Rodríguez et al., 2014*). Almost all *P. chihuahuana* populations are located on creeks or rivers and from north-east to north-west facing slopes (*Ledig et al., 2000*). *P. chihuahuana* is commonly associated with species of the genera *Pinus* and *Quercus,* and occasionally with species of the genera *Abies, Pseudotsuga, Cupressus, Populus, Juniperus,* and *Prunus* (*Gordon, 1968*; *Wehenkel et al., 2015*).

Previous studies have stated the hypothesis that populations of *P. chihuahuana* collapsed during the Holocene warming, with a significant reduction of their effective population size (*Jaramillo-Correa et al., 2006*). This rare species is economically unimportant as a timber tree; however, large mature trees were harvested sometime in the past (*Thomas & Farjon, 2013*). Grazing and forest fires have also contributed to reduce population sizes (*Ledig et al., 1997*). Currently, several other factors also threaten *P. chihuahuana* populations, including the low reproductive capacity resulting from high levels of self-fertilization and mating between closely related individuals (*Ledig et al., 1997*). Recent research revealed problems of genetic erosion in one population, San José de las Causas (*Wehenkel & Sáenz-Romero, 2012*).
Different approaches have been used to study *Picea chihuahuana* from the perspectives of ecology (*Narváez, 1984*; *Ledig et al., 2000*), genetic structure (*Ledig et al., 1997*; *Jaramillo-Correa et al., 2006*; *Wehenkel et al., 2012*; *Wehenkel & Sáenz-Romero, 2012*; *Quiñones Pérez, Sáenz-Romero & Wehenkel, 2014*; *Quiñones Pérez et al., 2014*; *Wehenkel, Sáenz-Romero & Jaramillo-Correa, 2015*), and climate change (*Ledig et al., 2010*).

In a recent study, *Mendoza-Maya et al. (2015)* suggested assisted migration as an alternative to the *ex situ* conservation of *P. chihuahuana*, taking into consideration the genetic structure and diversity of the populations and also predictions regarding the future climate of the habitat. However, detailed background information is required to enable development of plans for protecting and conserving species and in order to achieve successful assisted migration. Thus, it is important to identify differences between populations in relation to environmental conditions (*Aguilar-Soto et al., 2015*). The vitality and genetic diversity of populations, which affect vigor, evolution, and adaptability of the species, must also be considered (*Frankham, Ballou & Briscoe, 2002*; *Reed & Frankham, 2003*). In other words, genetic diversity is vital for increasing population fitness by reducing inbreeding depression in the short term and, in the longer term, to develop new local adaptations in response to environmental changes (*Reed & Frankham, 2003*). Genetic diversity also affects ecological processes such as primary productivity, population recovery from disturbances, interspecific competition, community structure, and fluxes of energy and nutrients (*Hughes et al., 2008*). AFLP markers (amplified fragment length polymorphism) can be used to describe genetic diversity (*Meudt & Clarke, 2007*). Outlier AFLP markers were found in several studies (e.g., *Nunes et al., 2012*), that were associated with different abiotic and biotic conditions (e.g., *Jump et al., 2006*; *Wehenkel, Corral-Rivas & Castellanos-Bocaz, 2010*).

In this study, we examined fourteen *P. chihuahuana* populations with the overall aim of discriminating the populations and clusters of this unique tree species. For this purpose we: (i) determined 74 variables: 22 climatic, 27 edaphic, 10 dasometric, four density variables and other two population variables, as well as six genetic variables and three species diversity indices were tested by using putatively neutral and adaptive AFLP markers, (ii) identified suitable variables for separating populations, and (iii) tested for correlation between genetic diversity, dasometric, and environmental factors. Our purpose was seeking for any significant differences, in order to predict species distribution by discriminant analysis; the results led us to make proposals for *ex situ* conservation of *P. chihuahuana*.

## MATERIALS AND METHODS

### Study area

The study was conducted in 14 populations of *P. chihuahuana* located in five municipalities of the state of Durango and two municipalities of Chihuahua, Mexico (Table 1 and Fig. 1). The 14 locations were selected in order to cover three geographically distinct clusters of the natural distribution along the species (north, center, and south). Each location was represented by one $50 \times 50$ m (0.25 ha) plot established in the center of the respective

**Table 1** Locations of the 14 *Picea chihuahuana* populations under study.

| Geographical Group | Code | Property | Municipality | Location | Population Size (T)* | Sample size | Latitude (N) | Longitude (W) | Altitude (m) |
|---|---|---|---|---|---|---|---|---|---|
| Northern | TN | El Ranchito | Bocoyna | La Tinaja | 99 | 50 | 27°57′27″ | 107°46′13″ | 2,380 |
| | RC | El Ranchito | Bocoyna | El Ranchito | 217 | 51 | 27°57′20″ | 107°45′12″ | 2,414 |
| | CV | El Ranchito | Bocoyna | El Cuervo | 140 | 50 | 27°57′01″ | 107°46′18″ | 2,500 |
| | TY | Los Volcanes | Bocoyna | Talayote | 291 | 53 | 27°55′03″ | 107°49′01″ | 2,355 |
| | TR | El Ranchito | Bocoyna | Las Trojas | 834 | 51 | 27°54′27″ | 107°45′17″ | 2,395 |
| | VN | San Javier | Bocoyna | El Venado | 1,785 | 57 | 27°45′41″ | 107°41′33″ | 2,311 |
| Central | LQ | El Caldillo y su anexo El Vergel | Balleza | La Quebrada | 877 | 50 | 26°28′13″ | 106°21′51″ | 2,730 |
| | PPR | Chiqueros | Guanaceví | Paraje Piedra Rayada | 3,564 | 42 | 26°09′15″ | 106°24′17″ | 2,600 |
| | QD | Chiqueros | Guanaceví | Quebrada de los Durán | 2,628 | 49 | 26°08′48″ | 106°22′53″ | 2,570 |
| | CB | Private property | Canelas | Cebollitas | 172 | 51 | 25°05′55″ | 106°26′27″ | 2,450 |
| Southern | SJ | San José de las Causas | San Dimas | San José de las Causas | 21 | 51 | 24°01′07″ | 105°47′56″ | 2,480 |
| | SB | El Brillante | Pueblo Nuevo | Santa Bárbara | 148 | 48 | 23°39′44″ | 105°26′20″ | 2,725 |
| | ACH | Santa Maria Magdalena de Taxicaringa | Mezquital | Arroyo del Chino | 46 | 17 | 23°21′05″ | 104°43′05″ | 2,600 |
| | LP | Santa Maria Magdalena de Taxicaringa | Mezquital | La Pista | 919 | 49 | 23°19′52″ | 104°45′00″ | 2,685 |

**Notes.**
*Taken from Table 6 of *Ledig et al. (2000).*

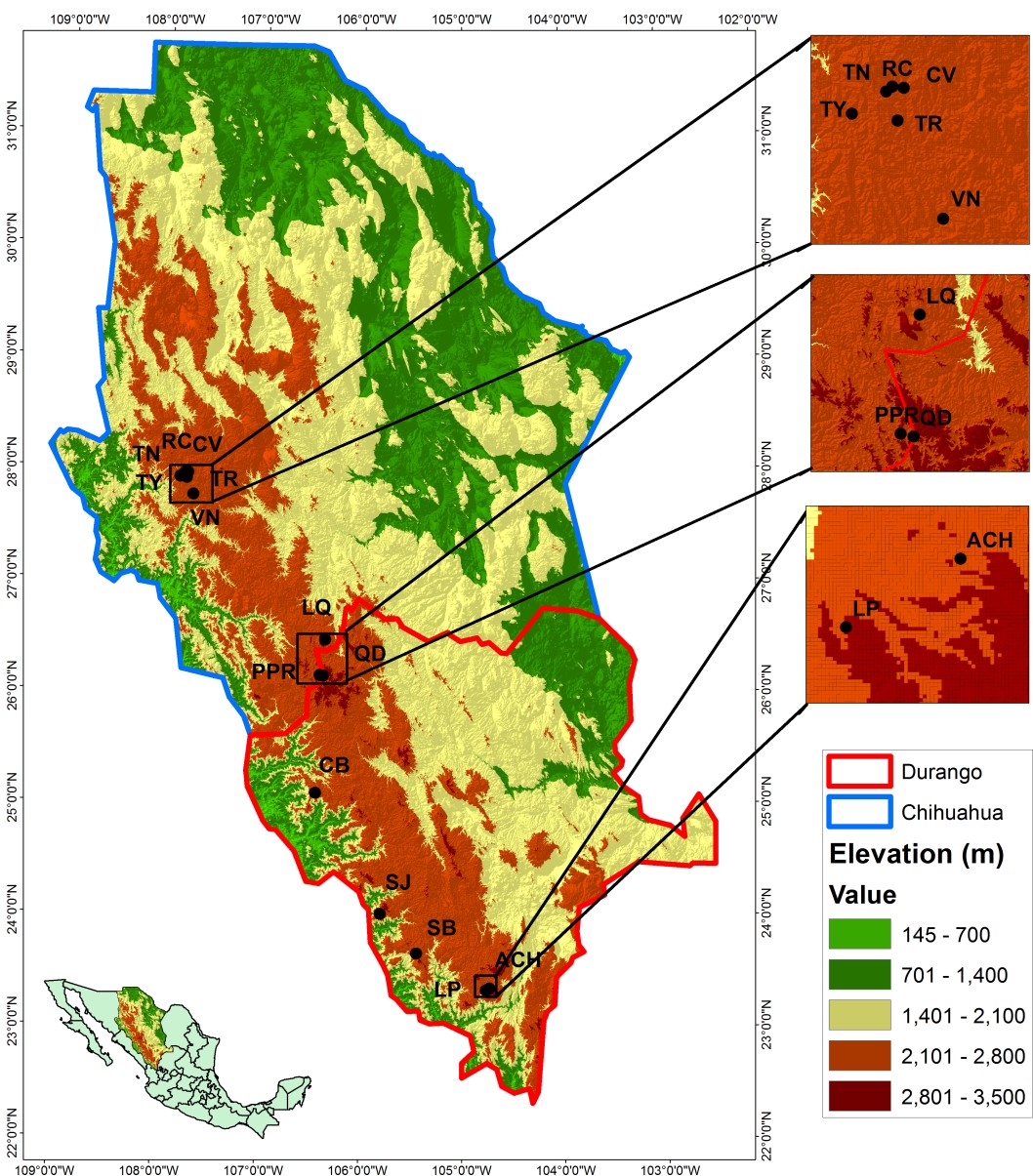

**Figure 1** Locations of the studied populations: La Tinaja (TN), El Ranchito (RC), El Cuervo (CV), Talayote (TY), Las Trojas (TR), El Venado (VN), La Quebrada (LQ), Paraje Piedra Rayada (PPR), Quebrada de los Duran (QD), Cebollitas (CB), San José de las Causas (SJ), Santa Bárbara (SB), Arroyo del Chino (ACH), La Pista (LP).

population. Following *Wehenkel et al. (2015)*, all trees with diameter at breast height (DBH) ≥7.5 cm were scored in regard to position, DBH, height, and species affiliation. Field experiments were approved by the Secretariat of Environment and Natural Resources, Mexico (SEMARNAT; permit number SGPA/DGVS/02835/12).

## Determination of climate variables

The climate model developed by *Rehfeldt (2006)*, based on thin plate spline (TPS) of *Hutchinson (1991)* and *Hutchinson (2004)*, and explained for its Mexican implementation
**Table 2  Descriptive statistics for the 22 physiographic and climatic variables, SD = standard deviation, $n = 14$.**

|  | Variable Climatic | Minimum | Maximum | Mean | SD |
|---|---|---|---|---|---|
| Long | Longitude (degrees) | −107.817 | −104.718 | −106.703 | 1.064 |
| Lat | Latitude (degrees) | 23.30 | 27.90 | 26.12 | 1.88 |
| Elev | Elevation (m) | 2,311 | 2,730 | 2,509 | 132 |
| Mat | Mean annual temperature (°C) | 9.70 | 11.90 | 10.80 | 0.50 |
| Map | Mean annual precipitation (mm) | 700 | 1,350 | 905.9 | 218.9 |
| Gsp | Growing season precipitation, April to September (mm) | 520 | 941 | 658.0 | 150.5 |
| Mtcm | Mean temperature in the coldest month (°C) | 3.80 | 7.30 | 5.00 | 1.00 |
| Mmin | Mean minimum temperature in the coldest month (°C) | −5.6 | −1.3 | −4.2 | 1.4 |
| Mtwm | Mean temperature in the warmest month (°C) | 13.80 | 17.20 | 15.80 | 1.08 |
| Mmax | Mean maximum temperature in the warmest month (°C) | 21.70 | 26.60 | 24.80 | 1.40 |
| Sday | Julian date of the last freezing date of spring | 1,260 | 163.00 | 151.00 | 10.0 |
| Fday | Julian date of the first freezing date of autumn | 266 | 295 | 281 | 9 |
| Ffp | Length of the frost-free period (days) | 104 | 165 | 134 | 19 |
| Dd5 | Degree-days above 5 °C | 1,873 | 2,593 | 2,275 | 178 |
| Gsdd5 | Degree-days above 5 °C in the frost-free period | 974 | 1,679 | 1,323 | 220 |
| D100 | Julian date the sum of degree-days above 5 °C reaches 100 | 35 | 69 | 56 | 10 |
| DD0 | Degree-days below 0 °C (based on mean monthly temperature) | 0 | 39 | 20 | 12 |
| Mmindd0 | Degree-days below 0 °C (based on mean minimum monthly temperature) | 427 | 907 | 780 | 157 |
| Smrpb | Summer precipitation balance: (Jul+Aug+Sep)/(Apr+May+Jun) (mm) | 3.83 | 4.96 | 4.47 | 0.36 |
| Smrsprpb | Summer/Spring precipitation balance: (Jul+Aug)/(Apr+May) (mm) | 10.53 | 14.48 | 12.49 | 1.00 |
| Sprp | Spring precipitation (Apr+May) (mm) | 26 | 43 | 32 | 6 |
| Smrp | Summer precipitation (Jul+Aug) (mm) | 316 | 544 | 396 | 81 |
| Winp | Winter precipitation (Nov+Dec+Jan+Feb) (mm) | 100 | 326 | 172 | 62 |

by *Sáenz-Romero et al. (2010)*, was used to estimate 22 climate variables in each population. This model yielded data from standardized monthly mean, minimum, and maximum values of temperature and precipitation from more than 200 climate stations in Chihuahua and Durango, for the period 1961–1990. Point estimates of climate measures were obtained from a national database managed by the University of Idaho (http://forest.moscowfsl.wsu.edu/climate/), for which the geographical coordinates (latitude, longitude, and elevation) are required as input data to interrogate the climate splines. Estimation of variables included: mean annual precipitation (mm), mean temperature in the warmest month (°C), mean maximum temperature in the warmest month (°C), Julian date of the first freezing date of autumn, and precipitation during the growing season (April–September) (mm) (Table 2).

## Determination of edaphic variables

In each location, a soil subsample (250 g) was collected at a depth of 0–15 cm at the base of the stems of four *Picea chihuahuana* trees. The four soil subsamples were combined to make a 1,000 g sample per population (14 samples in total) for analysis of 27 edaphic variables: the texture (relative proportion of sand, silt, and clay), density (Den) (g/cm$^3$), concentration of calcium carbonate ($CaCO_3$) (%), pH ($CaCl_2$, 0.01 M), concentrations of potassium (K)(ppm), magnesium (Mg) (ppm), sodium (Na) (ppm), copper (Cu) (ppm), iron (Fe) (ppm), manganese (Mn) (ppm), zinc (Zn) (ppm), and calcium (Ca) (ppm) in the soil were determined by the methods described by *Castellanos, Uvalle-Bueno & Aguilar-Santelises (1999)*. Phosphorus (P) (ppm) was determined by the method of *Olsen et al. (1954)*. Nitrate ($NO_3$) (kg /ha) was determined by the method of *Baker (1967)* and the relative organic matter (%) (OM) contents were determined by the method of *León & Aguilar (1987)*. Electrical conductivity (CE) (dS/m) was determined by the method described by *Vázquez & Bautista (1993)*. Finally, the cation exchange capacity (meq 100 g soil) (CEC) and the relative proportions (%) of hydrogen, Ca, M, K, Na and other bases (o.b.) in the CEC were estimated on the basis of the Ammonium Acetate Method (pH 8.5). The hydraulic conductivity (HC) (cm/h) was determined by the method of *Mualem (1976)* and percent saturation (Sat) (%) was estimated by the method of *Herbert (1992)*. Edaphic variables are described in Table 3.

## Determination of dasometric, density and population variables

For each of the 14 plots we estimated the individual diameter at breast height (DBH), basal area (G), height (H), maximum diameter at breast height ($DBH_{max}$), maximum height ($H_{max}$) of *P. chihuahuana* trees. For each plot we also estimated the following variables considering together all tree species found per plot (see details in *Wehenkel et al., 2015*): individual total diameter at breast height ($DBH_{tot}$) and individual total height ($H_{tot}$). Besides we registered the total maximum diameter at breast height considering together all tree species per plot ($DBH_{max,tot}$) and total maximum height for all tree species found per plot ($H_{max,tot}$), according to *Assmann (1970)*. We also estimated the total number of individuals of *P. chihuahuana* per plot (N), quadratic DBH of *P. chihuahuana* per plot ($D_g$), total number of individuals per plot ($N_{tot}$), basal area per plot ($G_{tot}$) and quadratic DBH per plot ($D_{g,tot}$), according to *Wehenkel et al. (2015)* (Table 4). Two other population variables were considered: population size (T) and geographical distance between neighbor populations ($d_{min}$). T was taken from Table 6 of *Ledig et al. (2000)*. $d_{min}$ was calculated by GenAlex 6.5 (*Peakall & Smouse, 2006*) (Table 1). All the 40 known populations, based on their geographical coordinates (Table 1), were included for the distance calculations.

## Determination of genetic diversity variables

Needles were sampled from 669 individuals (seedlings, saplings and trees) of *P. chihuahuana* in the 14 populations (plots) studied (i.e., 17–57 individuals per plot), for determination of genetic diversity variables (Table 5).

The DNA was extracted using the DNeasy 96 Plant Kit (QIAGEN, Hilden, Germany). The amplified fragment length polymorphism (AFLP) analysis was conducted according

**Table 3 Descriptive statistics for the 27 soil variables, SD = standard deviation, $n = 14$.**

| | Soil variable | Minimum | Maximum | Mean | SD |
|---|---|---|---|---|---|
| EC | Electric conductivity (dS/m) | 0.24 | 2.19 | 0.82 | 0.51 |
| $NO_3$ | Nitrate (kg/ha) | 14.78 | 564.69 | 179.61 | 137.0 |
| P | Phosphorus (ppm) | 6.88 | 114.68 | 27.03 | 33.24 |
| OM | Organic material (%) | 3.35 | 17.49 | 9.33 | 4.65 |
| %$CaCO_3$ | Calcium carbonate (%) | 0.36 | 12.56 | 2.09 | 3.81 |
| %Sat | Percent saturation (%) | 29.00 | 92.00 | 66.64 | 17.27 |
| %Sand | Sand (%) | 51.26 | 75.26 | 64.26 | 7.64 |
| %Silt | Silt (%) | 15.28 | 33.28 | 23.99 | 5.64 |
| %Clay | Clay (%) | 7.46 | 17.46 | 11.75 | 3.02 |
| Den | Density (g/cm$^3$) | 0.70 | 1.07 | 0.89 | 0.13 |
| pH | pH | 4.80 | 7.22 | 5.80 | 0.52 |
| Ca | Calcium (ppm) | 5.44 | 6.15 | 5.97 | 0.22 |
| Mg | Magnesium (ppm) | 2,340.00 | 6,090.00 | 4,147.71 | 1,086.96 |
| Na | Sodium (ppm) | 144.00 | 942.00 | 394.29 | 187.87 |
| K | Potassium (ppm) | 40.00 | 177.50 | 77.54 | 34.27 |
| Fe | Iron (ppm) | 191.00 | 6,225.00 | 1,697.18 | 1,587.93 |
| Zn | Zinc (ppm) | 31.28 | 313.72 | 142.81 | 72.22 |
| Mn | Manganese (ppm) | 0.32 | 12.56 | 4.69 | 4.23 |
| Cu | Copper (ppm) | 16.64 | 266.20 | 92.48 | 70.67 |
| %o.b. | Rel. proportion of other bases in CEC (%) | 0.16 | 1.04 | 0.45 | 0.26 |
| %Ca | Rel. proportion of Ca in CEC (%) | 4.22 | 7.09 | 5.80 | 0.69 |
| %Mg | Rel. proportion of Mg in CEC (%) | 41.87 | 69.52 | 56.38 | 7.98 |
| %K | Rel. proportion of K in CEC (%) | 5.29 | 15.06 | 8.69 | 2.62 |
| %Na | Rel. proportion of Na in CEC (%) | 2.36 | 21.95 | 10.01 | 6.04 |
| %H | Rel. proportion of H in CEC (%) | 0.34 | 1.76 | 0.93 | 0.33 |
| CEC | Cation exchange capacity (meq/100 g soil) | 15.30 | 33.00 | 18.18 | 7.19 |
| HC | Hydraulic conductivity (cm/h) | 20.77 | 72.72 | 37.76 | 13.23 |

to a modified version of the protocol of *Vos et al. (1995)*, described by *Simental-Rodríguez et al. (2014)*. The restriction enzymes used were Eco RI (selective primer: 5′-GACTGC GTACCAATTCNNN-3′) and Mse I (selective primer: 5′-GATGAGTCCTGAGTAANNN-3′). The primer combination E01/M03 (EcoRI-A/MseI-G) was used in the pre-AFLP amplification. Selective amplification was carried out with the fluorescent-labelled (FAM) primer pair E35 (EcoRI-ACA) and M70 (MseI-GCT). The AFLP products were separated in an ABI 3100 Genetic Analyzer, along with the GeneScan 500 ROX internal lane size standard (Applied Biosystems, Foster City, California, USA). Selection of the amplified restriction products was totally automated, and only strong and high quality fragments were considered. The size of the AFLP fragments was determined with the GeneScan® 3.7 and Genotyper® 3.7 software packages (Applied Biosystems, Foster City, California, USA). Binary AFLP matrices were created from the presence (code 1) or absence (code 0) at probable fragment positions. The quality and reproducibility of the analysis were verified according to *Ávila-Flores et al. (2016)*.
**Table 4** Descriptive statistics for 10 dasometric variables, four density variables and other population variables. Dasometric variables including all trees with diameter at breast height ≥ 7.5 cm, SD = standard deviation, $n = 14$.

| | | Minimum | Maximum | Mean | SD |
|---|---|---|---|---|---|
| **Dasometric variable** | | | | | |
| Dg | Quadratic diameter at breast height per plot (cm) | 0 | 40 | 30 | 11 |
| DBH | Diameter at breast height per plot (cm) | 0 | 35 | 27 | 9 |
| H | Height per plot (m) | 0.0 | 21.1 | 15.9 | 5.2 |
| $DBH_{max,}$ | Maximum diameter at breast height per plot (m) | 0 | 78 | 55 | 20 |
| $H_{max,}$ | Maximum height per plot (m) | 0.0 | 46.0 | 30.8 | 10.8 |
| $Dg_{tot}$ | Total Quadratic diameter (cm) per plot | 22 | 37 | 28 | 4 |
| $DBH_{tot}$ | Total diameter (cm) per plot | 18 | 33 | 24 | 3 |
| $H_{tot}$ | Total height (m) per plot | 9.7 | 17.9 | 14.1 | 2.1 |
| $DBH_{max,tot}$ | Total maximum diameter at breast height (cm) per plot | 55 | 104 | 75 | 15 |
| $H_{max,tot}$ | Total maximum height (m) per plot | 23.3 | 48.0 | 34.9 | 7.0 |
| **Density variable** | | | | | |
| N | Number of individuals per plot | 0 | 140 | 76 | 42 |
| G | Tree basal area per plot of (m²/ha) | 0.00 | 14.3 | 6.81 | 4.66 |
| $N_{tot}$ | Total number of individuals per plot | 152 | 736 | 370 | 139 |
| $G_{tot}$ | Total tree basal area (m²/ha) per plot | 13.70 | 53.28 | 22.41 | 9.69 |
| **Other population variables** | | | | | |
| $d_{min}$ | Geographical distance between neighbor populations (m) | 63 | 77,303 | 14,737 | 24,612 |
| T | Population size (tree number per population) | 21 | 3,564 | 951 | 1,264 |

**Table 5** Descriptive statistics for the nine genetic and species diversity variables, SD = standard deviation, $n = 14$.

| | Diversity variable | Minimum | Maximum | Mean | SD |
|---|---|---|---|---|---|
| $v_2$ | Mean genetic diversity | 1.43 | 1.60 | 1.52 | 0.06 |
| POLY | Percentage polymorphism | 0.80 | 1.02 | 0.94 | 0.07 |
| DW | Modified frequency-down-weighted marker value | 0.08 | 0.12 | 0.1 | 0.01 |
| $v_{2(adaptiveAFLP)}$ | Mean genetic diversity per outlier AFLP | 1.07 | 1.78 | 1.46 | 0.27 |
| $POLY_{(adaptiveAFLP)}$ | Percentage polymorphism per outlier AFLP | 0.26 | 1.02 | 0.78 | 0.31 |
| $DW_{(adaptiveAFLP)}$ | Modified frequency-down-weighted marker value per outlier AFLP | 0.002 | 0.02 | 0.01 | 0.07 |
| $v_{sp,0}$ | Species richness | 4.00 | 9.00 | 6.17 | 1.49 |
| $v_{sp,2}$ | Effective number of tree species | 1.92 | 4.46 | 3.39 | 0.80 |
| $v_{sp,inf}$ | Number of prevalent tree species | 1.49 | 3.00 | 2.31 | 0.46 |

The AFLP data were used to calculate three genetic diversity indices (Table 5): the modified frequency-down-weighted marker value (DW), the polymorphism percentage (POLY) (*Schönswetter & Tribsch, 2005*), and, the mean genetic diversity ($v_2$) *Gregorius (1978)*,

$$V_{2,j} = \left(\frac{1}{N}\right) \times \sum \left(\frac{1}{\sum p_{ij}^2}\right)$$

where: $p_{ij}$ is the relative frequency of a variant from the $i$ to the $j$ locus and $N$ is the sample number.

The value of DW is expected to be high when rare AFLPs are accumulated (*Schönswetter & Tribsch, 2005*). In order to equalize dissimilar sample sizes, the values of the three diversity indices were multiplied by a correction term ($N/(N-1)$), (*Gregorius, 1978*).

The values of these three genetic diversity indices were also calculated for putatively adaptive AFLP markers under natural selection (adaptive AFLP), detected in *P. chihuahuana* by *Simental-Rodríguez et al. (2014)*.

The values of tree species richness ($\nu_{sp,0}$), Simpson index ($\nu_{sp,2}$), and number of prevalent tree species ($\nu_{sp,inf}$) in the 14 plots were taken from *Simental-Rodríguez et al. (2014)* who used the same sampling strategy as in the present study (Table 5).

## Cluster analysis

First, in order to detect the optimal cluster set for population conditions which were almost homogeneous inside each cluster, but clearly different from any other clusters, we used the recent Affinity Propagation (AP) clustering technique, with the input preference to the 0 quantile ($q$) of the input similarities (*Bodenhofer, Kothmeier & Hochreiter, 2011*), along with the $k$-means clustering algorithm ($k$-means) (*Hartigan & Wong, 1979*). We also utilized the Calinski-Harabasz criterion (CHC) to determine the optimal number of clusters. CHC minimizes the within-cluster sum of squares and maximizes the between-cluster sum of squares. Therefore, the highest CHC value is related to the optimal set (of most compact clusters). The optimal set can be identified by a peak or at least an abrupt elbow on the linear plot of CHC values (*Legendre & Legendre, 1998*).

By contrast to the $k$-means, the conceptually new AP simultaneously includes all data points as potential exemplars. Furthermore, AP has several advantages over related techniques, such as $k$-centres clustering, the expectation maximization (EM) algorithm, Markov chain Monte Carlo procedures, hierarchical clustering and spectral clustering (see details in *Frey & Dueck, 2007*). More importantly, it does not need a pre-defined number of groups (*Bodenhofer, Kothmeier & Hochreiter, 2011*).

For all the *P. chihuahuana* populations, both the AP ($q = 0$) technique and the $k$-means clustering along with CHC were firstly applied to all the 74 predictor variables together, and then separately for the 22 climate variables, 27 soil variables, nine genetic and species diversity variables, 10 dasometric variables, four density variables, T and $d_{min}$ (Tables 2–5).

All analyses were implemented using the R Script for $k$-Means Cluster Analysis and "apcluster" software packages (*Bodenhofer, Kothmeier & Hochreiter, 2011*) executed in the R free statistical application (*R Core Team, 2015*).

The AP and $k$-means clustering techniques recommended only two clusters of *P. chihuahuana* populations under study, which were completely separated from each other by the latitude and several other predictor variables.

## Principal component analysis and logistic regression

Stepwise binomial multivariate logistic regression was used, which accepts independent variables even with heteroscedasticity and without a multivariate normal distribution (*Hosmer, Lemeshow & Sturdivant, 2013*). This regression tested for significant differences in climatic, dasometric, soil, genetic and species diversity variables between the southern

populations (value zero) and the northern (value one) populations of *P. chihuahuana* (Table 1). The R software (version 3.3.2) was used to conduct the analysis. A linear discrimination analysis (*Fisher, 1936*) was not applied, since not every independent variable was normally distributed.

From the 74 predictor variables in Tables 1–5 only those that were not highly correlated with other predictor variables were included, because logistic regression requires each variable to be independent from each other (i.e., little or no multicollinearity). These predictor variables were found applying a varimax-rotated Principal Component Analysis (PCA) (*Pearson, 1901*). Therefore, only one variable from each PCA factor and with the highest factorial loads was selected for logistic regression.

Variables were excluded from the models if the probability of incorrectness (*p*) was greater than or equal to 5%. Stepwise selection (forward and backward) was performed to select the most informative variables for inclusion in the models. This procedure was done using the glm (*generalized linear model*) (family = "binomial"), the step AIC (Akaike information criterion) function and the exact AIC using the "MASS" package (*Venables & Ripley, 2002*) in R (*R Core Team, 2015*). The AIC, standard error (SE) and residual deviance were used to evaluate the goodness-of-fit.

## Ordinary kriging analysis

Ordinary kriging (ordinary Gaussian process regression model) was used to illustrate the spatial distribution of genetic diversities ($v_2$, POLY, and DW) in *P. chihuahuana* (*Batista et al., 2016*). The mathematical models for describing the semivariance were: the spherical model, exponential model, Gaussian model, and Stein's parameterization. The best interpolation model was detected using 10-fold cross validation point-by-point. Correlation between the observed and predicted values ($r_k$) and the Unbiased Root Mean Squared Error of the residual (URMSE) were used to assess the goodness-of-fit. Finally, the model with the best fit was selected to create the prediction surface map of genetic diversity.

This modeling was realized using the CRS, SpatialPixelsDataFrame, autoKrige, autoKrige.cv, and compare.cv functions and using the "SP" (*Pebesma & Bivand, 2005*) and "automap" packages (*Hiemstra et al., 2009*) in R (*R Core Team, 2015*).

## Spearman correlations

Spearman's correlation ($r_s$) test (*Hauke & Kossowski, 2011*) was used to analyze the relationships between genetic diversity and the climatic, soil, dasometric variables, $d_{min}$ and T. The test was implemented using R 3.2.3 statistical software (*R Core Team, 2015*). A Bonferroni correction was applied to calculate the new critical significance level ($\alpha^* = 0.00023$), by dividing the proposed critical significance level ($\alpha = 0.05$) by the number of comparisons ($m = 213$) (*Hochberg, 1988*).

## RESULTS

### Cluster analysis

The Affinity Propagation clustering technique and the *k*-means clustering algorithm recommended two clusters based on the 74 predictor variables; the same grouping was
found by using only the 22 climate variables under study (Fig. 2). The first cluster included the nine most northern *P. chihuahuana* populations under study (TN, RC, CV, TY, TR, VN, LQ, PPR and QD). While the second group comprised the five most southern populations (CB, SJ, SB, ACH and LP) (Table 1, Fig. 1). A cluster analyses was also applied with respect to the 27 soil variables, six genetic diversity indices, three species diversity indices and 14 dasometric variables, but patterns related to the geographical coordinates (i.e., latitude and longitude) were not found.

## Principal component analysis and logistic regression

Eight uncorrelated variables (Mmin, Gsdd5, $v_{sp,0}$, $Dg_{tot}$, $NO_3$, Zn, %Mg and $H_{max,tot}$) from the 14 *P. chihuahuana* populations were selected for logistic regression analysis. This selection was based on a PCA (Fig. 3). The logistic regression analysis revealed that the Mmin clearly separated the southern from northern populations (Fig. 4A).

However, Mmin is a variable from the PCA factor group 1 (F1), and was strongly correlated with other eight F1 variables with high factorial loads (Long, Map, Gsp, Mtcm, Mmax, Mmindd0, Smrp, and %Sand), indicating that these eight variables were also important for characterizing and separating the two clusters. Since these eight variables characterized to 100% the two clusters, we considered that the binominal logit models were no longer needed.

Moreover, the probability ($P$) of being a northern population is higher if the sand proportion in the soil (%Sand) was significant lower (SE of Intercept = 10.178, $p = 0.0242$, SE of %Sand = 0.146, $p = 0.0199$, residual deviance: 9.467 on 12 degrees of freedom, AIC: 13.467) (Fig. 4B). The model is:

$$P = \frac{1}{1 + e^{-0.3396\ Sand + 22.937}}. \tag{1}$$

Significant differences in genetic variables and species diversity between southern and northern populations and locations were not found, although higher $v_2$ and DW were more probable in the northern populations.

According to the most important variables for the separation of the two clusters (Tables S1–S8) the logistic regression analysis of the *P. chihuahuana* populations revealed that the southern locations were characterized by more abundant precipitation in the summer, in the growing season and in the annual average in comparison to the northern locations. The southern populations also showed higher mean temperature in the coldest month, lower mean maximum temperature in the warmest month and less degree-days below 0 °C (based on mean minimum monthly temperature).

## Ordinary kriging analysis and Spearman correlations

There was not a genetic diversity gradient from the northern to the southern cluster. The best kriging model was found for $v_2$ using the exponential model ($r_k = 0.842$; URMSE = 0.019) (Fig. 5A). On the other hand, the goodness-of-fit of both the PLOY and DW models was poorer, respectively ($r_k = 0.633$, URMSE = 0.063 and $r_k = 0.4165$, URMSE = 0.123). The prediction and standard error surface maps of $v_2$ are shown in Fig. 5B.

After Bonferroni correction, the mean genetic diversity $v_2$ of *P. chihuahuana* was significantly correlated with the mean temperature in the warmest month (°C) (Mtwm)

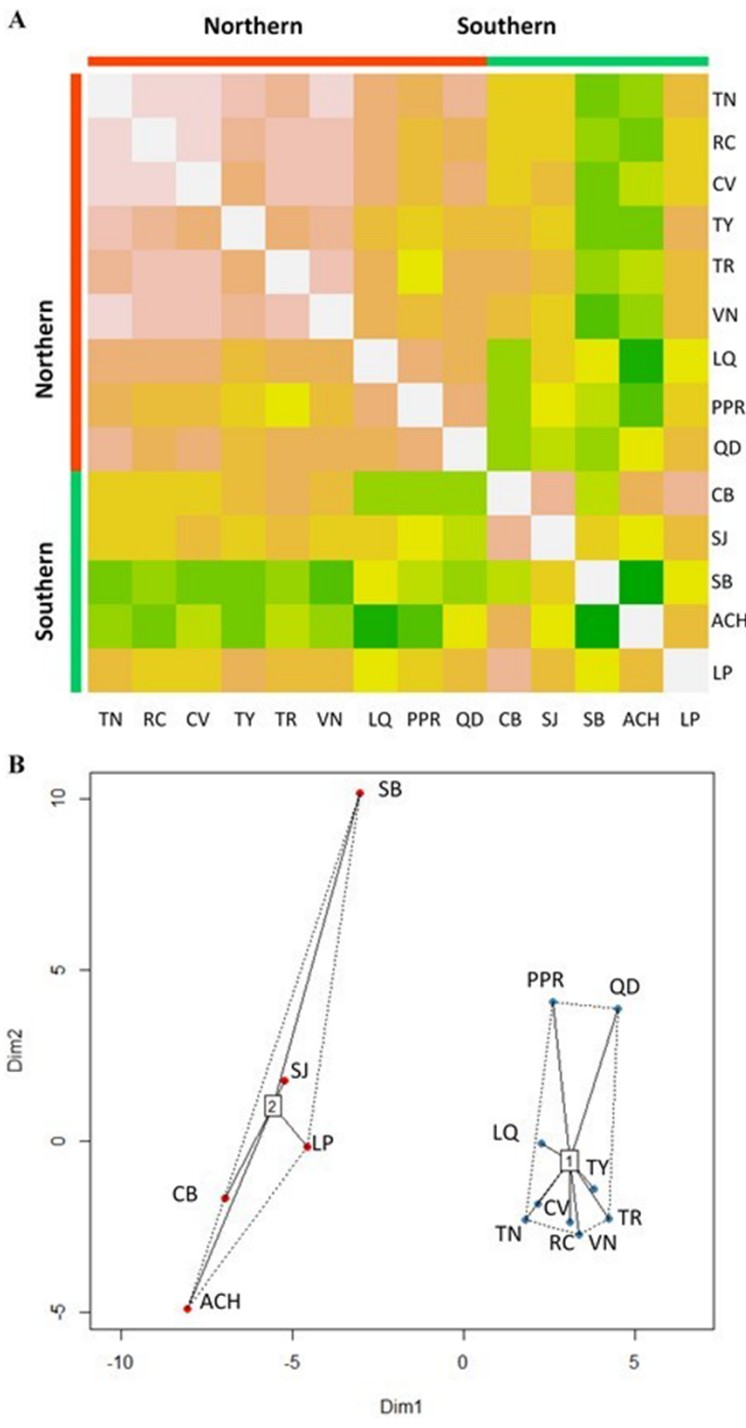

**Figure 2 Clusters of *Picea chihuahuana* populations.** (A) Heat map to visualize the data and to identify clusters of similar environmental conditions based on the Affinity Propagation (AP) clustering, with the quantile = 0. This analysis indicates that the 14 populations can be subdivided into two clusters displayed as bright orange squares across the diagoal. Northern populations: La Tinaja (TN), El Ranchito (RC), El Cuervo (CV), Talayote (TY), Las Trojas (TR), El Venado (VN), La Quebrada (LQ), Paraje Piedra Rayada (PPR), Quebrada de los Duran (QD). Southern populations: Cebollitas (CB), San José de las Causas (SJ), Santa Bárbara (SB), Arroyo del Chino (ACH), P masyúscula La Pista (LP) (B) Clusters based on *k*-means. Cluster 1 includes the nine northern populations; Cluster 2 includes the five most southern populations.

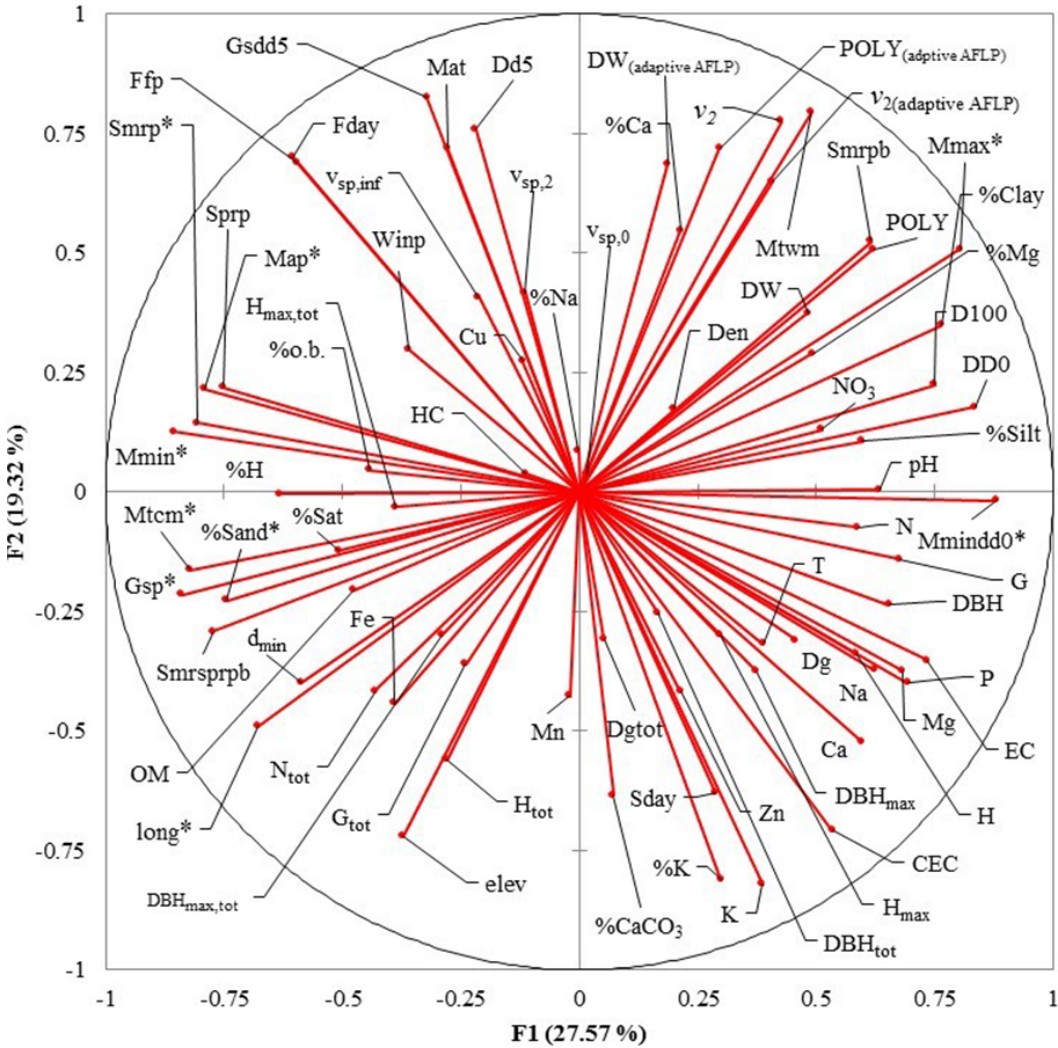

**Figure 3 Principal Component Analysis (PCA) plot of variables under study in the factor groups F1 and F2.** The most important variables to separate the southern from northern populations of *Picea chihuahuana* are asterisked. Abbreviations are defined in Table 2–Table 5.

($p = 0.0002$) (Table 6, Fig. 6). Genetic diversity of *P. chihuahuana* calculated with putatively adaptive AFLP markers was not statistically significantly correlated with any environmental factor. Finally, no significant positive correlations were observed between any of the three genetic diversity indices and population size. The negative association between genetic diversity and geographical distance to the next population was not significant ($r_s(v_2 \times d_{min}) = -0.46$, $p = 0.09$; $r_s(\text{PLOY} \times d_{min}) = -0.42$, $p = 0.13$; $r_s(\text{DW} \times d_{min}) = -0.24$, $p = 0.41$).

## DISCUSSION

Our main findings show that the southern and northern *P. chihuahuana* populations are characterized by different climate conditions. Seven climate variables, besides the geographical longitude and the sand proportion in soil (Fig. 3, Tables S1–S8) were identified

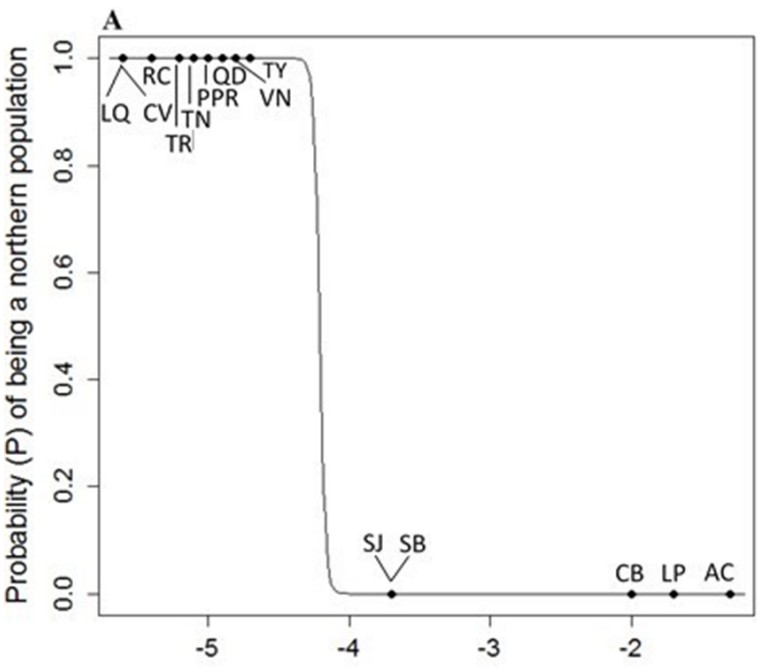

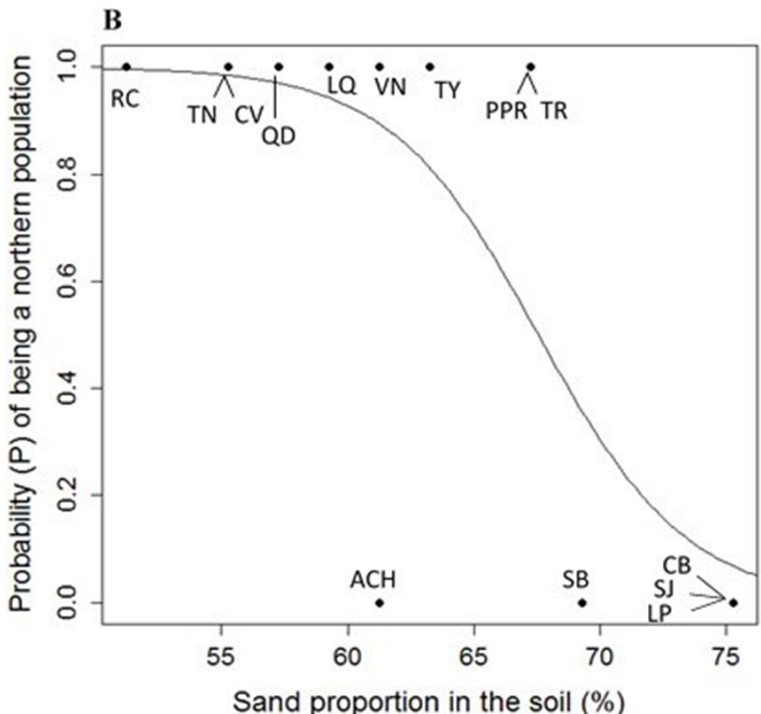

**Figure 4** (A) Logistic model between mean minimum temperature in the coldest month (Mmin) vs. probability (*P*) of being a northern population of *Picea chihuahuana*. (B) Logistic model between mean sand proportion (%) vs. probability (*P*) of being a northern population of *Picea chihuahuana*. Abbreviations are defined in Table 1 and Fig. 1.
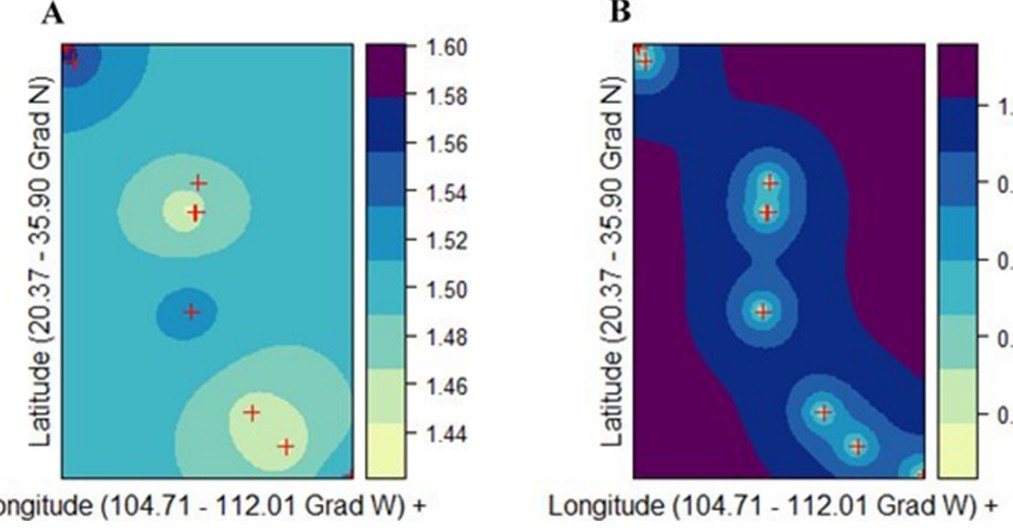

**Figure 5** Ordinary kriging analysis of the spatial genetic distribution of genetic diversity $v_2$ in *Picea chihuahuana* based on 14 populations studied (marked with red crosses); (A) Kriging prediction (correlation between the observed and predicted values equals 0.84), $v_2$ values shown on the right-hand side, and (B) Kriging standard error, error values shown on the right-hand side. The exponential model was the best mathematical model for describing the semivariance.

**Table 6** Correlation between genetic diversity ($v_2$) and climate and soil variables in 14 *Picea chihuahuana* populations.

| | Genetic diversity ($v_2$) | |
|---|---|---|
| | *Spearman r* | *P* |
| Long | −0.74 | 0.0027 |
| Mtwm | 0.83 | 0.0002* |
| Mmax | 0.70 | 0.0058 |
| D100 | 0.68 | 0.0074 |
| DD0 | 0.67 | 0.0088 |
| Smrpb | 0.69 | 0.0061 |
| Smrsprpb | −0.68 | 0.0076 |
| Clay | 0.67 | 0.0091 |

**Notes.**
*Significant after Bonferroni correction.

as important and relevant for separating the two groups and explained almost 100% of the variability. However, the most important climate variable to differentiate the *P. chihuahuana* populations was Mmin, which is strongly correlated with Long, Map, Gsp, Mtcm, Mmax, Mmindd0, Smrp, and %Sand. These findings are consistent with other studies, since several authors have reported that the distribution of species and populations depends on both climate and soil (e.g., *Condit et al., 2013*; *Toledo et al., 2012*; *John et al., 2007*).

The southern locations were characterized by more oceanic climate, probably caused by absence of the mountain barrier of Baja California peninsula, northwestern Mexico. The maximum temperatures in the northern locations of the *P. chihuahuana* populations were
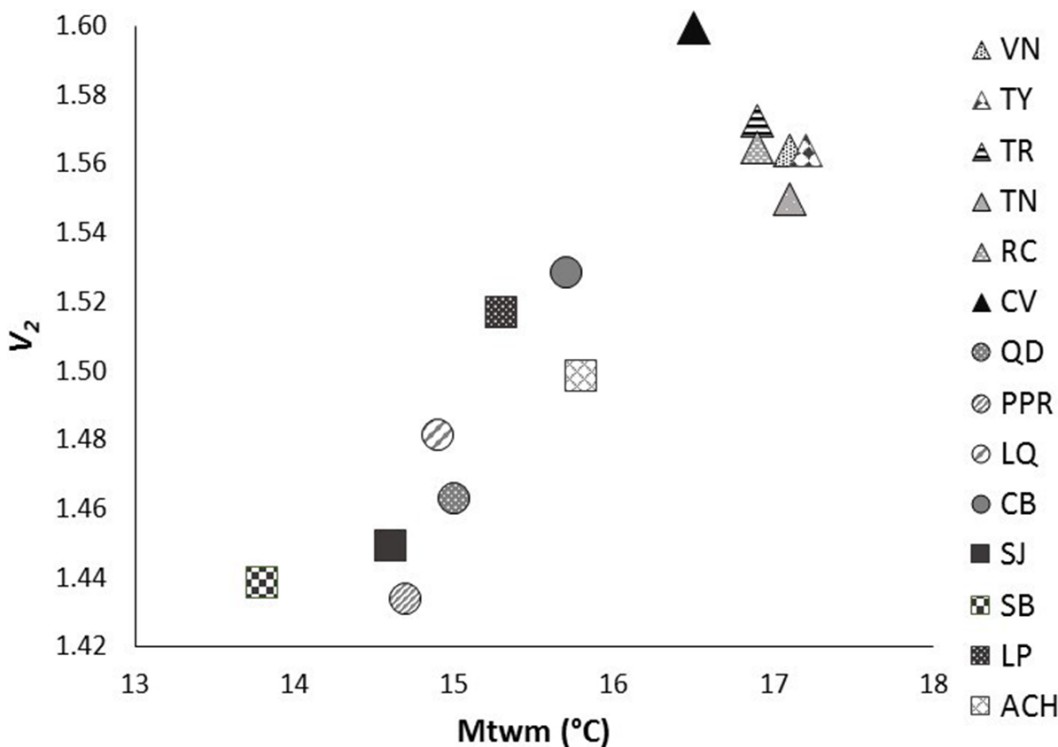

**Figure 6** Significant relationship between genetic diversity ($v_2$) and mean temperature in the warmest month (Mtwm) in the 14 studied populations of *Picea chihuahuana* after Bonferroni correction: La Tinaja (TN), El Ranchito (RC), El Cuervo (CV), Talayote (TY), Las Trojas (TR), El Venado (VN), La Quebrada (LQ), Paraje Piedra Rayada (PPR), Quebrada de los Duran (QD), Cebollitas (CB), San José de las Causas (SJ), Santa Bárbara (SB), Arroyo del Chino (ACH), La Pista (LP). Triangles represent the geographically northern cluster. Circles represent the geographically central cluster. Squares represent the geographically southern cluster.

also higher than in the southern ones. However, the future climate conditions, i.e., likely even higher temperatures and less precipitation may strongly restrict biomass production and the vitality of the most northern populations. This was observed by *Ledig et al. (2010)* who identified the most northern locations as the first group that may be threatened with extinction in some climate change projections.

The genetic diversity in *P. chihuahuana* is mostly moderate compared with other *Picea* species (*Simental-Rodríguez et al., 2014*; *Wehenkel & Sáenz-Romero, 2012*). The genetic diversity across all the AFLPs studied was not an important variable for separating the two clusters of *P. chihuahuana* populations (Fig. 5). However, it was significantly correlated with Mtwm (Table 6 and Fig. 6), where sites with warmer Mtwm harbor populations with larger genetic diversity. The most northern populations in the municipality of Bocoyna, Chihuahua were the sites with the highest Mtwm and aridity (lower precipitation values) (Tables S1–S8).

The genetic diversity among the putatively adaptive AFLPs was not significantly related to other variables. The relationships observed were probably not determined by adaptation, but by differences in the degree of isolation, which could influence gene flow and genetic

drift (*Ledig et al., 1997*; *Jaramillo-Correa et al., 2006*; *Quiñones Pérez, Sáenz-Romero & Wehenkel, 2014*). In comparison to the center and south, the most northern populations (municipality of Bocoyna) were much closer. After considering together the 11 documented populations in the Municipality of Bocoyna, Chih., the separation distances were: minimum 0.1 km, mean 13 km and maximum 25 km to each other. The fact that northern populations are located closer to each other may directly lead to a greater genetic exchange and a lower tendency for genetic drift and inbreeding and thus, to a higher level of genetic diversity (*Hamrick, Godt & Sherman-Broyles, 1992*; *Ledig et al., 1997*). This assumption was confirmed by the negative, but not significant association between genetic diversity and geographical distance between neighbor populations detected in our study.

*Jaramillo-Correa et al. (2006)* also found that the diversity of cpDNA in *P. chihuahuana* decreased from northern to southern areas (with the highest to the lowest Mtwm, respectively). These authors assumed that genetic drift, rather than selection, was the main factor determining the population diversity in the Chihuahua spruce. Moreover, the observations of *Ledig et al. (1997)*, based on isozyme analysis, also suggest the importance of drift and inbreeding in the recent evolution of this tree species.

Measurement of these environmental variables may be useful to identifying suitable and similar sites to those where the original stands are still growing, which may help to improve reforestation success. However, it will be important to specifically consider local micro climatic conditions that are not easy modelled with simple macro climate models (*Aguilar-Soto et al., 2015*), but can be recorded at new local weather stations within the populations.

## CONCLUSIONS

Our findings have three important practical implications in relation to *ex situ* conservation: first, at least two different groups (clusters of natural populations) of *P. chihuahuana* exist (according to the results of our cluster analysis), as local adaptation takes place because of the different climate and soil conditions. Climate has been recognized as the main driver of adaptation (*Vander, Bischoff & Smith, 2010*). These different groups are also designated by genetic differences between the southern and northern populations (*Ledig et al., 1997*; *Jaramillo-Correa et al., 2006*; *Quiñones Pérez, Sáenz-Romero & Wehenkel, 2014*), even if only most likely using neutral markers. Therefore, future reforestation programs should only be established with seed sources from the same geographical group. Second, there are not relevant climate environmental and genetic differences within each of the two clusters. Thus, seed from different populations of the same group could be mixed for improvement of genetic diversity levels. Third and finally, this study revealed the special macro-climate and soil conditions needed in the locations where *P. chihuahuana* is growing. Therefore, knowledge of these special conditions may be very helpful to find adequate reforestation locations in Mexico and other countries, which should have similar characteristics to the original sites.

## ACKNOWLEDGEMENTS

We are grateful to the two anonymous reviewers for their careful reading of the manuscript and their suggestions to improve it.

### Funding

This study was supported by joint funding from the Mexican Council of Science and Technology (CONACyT) and the Ministry of Education (SEP; Project CB-2010-01 158054). The funders had no role in study design, data collection and analysis, decision to publish, or preparation of the manuscript.

### Grant Disclosures

The following grant information was disclosed by the authors:
Mexican Council of Science and Technology (CONACyT).
Ministry of Education: SEP; Project CB-2010-01158054.

### Competing Interests

The authors declare there are no competing interests.

### Author Contributions

- Iliana Karina Dominguez-Guerrero analyzed the data, contributed reagents/materials/analysis tools, wrote the paper, prepared figures and/or tables, data sampling.
- Samantha del Rocío Mariscal-Lucero analyzed the data, wrote the paper, prepared figures and/or tables.
- José Ciro Hernández-Díaz, Berthold Heinze and José Ángel Prieto-Ruiz reviewed drafts of the paper.
- Christian Wehenkel conceived and designed the experiments, performed the experiments, analyzed the data, contributed reagents/materials/analysis tools, wrote the paper, prepared figures and/or tables, reviewed drafts of the paper, data sampling.

### Field Study Permissions

The following information was supplied relating to field study approvals (i.e., approving body and any reference numbers):

Field experiments were approved by SEMARNAT, Mexico (approval numbers: SGPA/DGVS/02578/12, SGPA/DGVS/05688/16, SGPA/DGVS/07916/16).

### Data Availability

The raw data has been supplied as a Supplementary File.

### Supplemental Information

Supplemental information for this article can be found online at http://dx.doi.org/10.7717/peerj.3452#supplemental-information.

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
