# Peer review of "Discrimination of Picea chihuahuana Martinez populations on the basis of climatic, edaphic, dendrometric, genetic and population traits"

_PeerJ, doi:10.7717/peerj.3452_

## Round 0.1 · original submission · Major Revisions

· Academic Editor

Major Revisions

Both reviewers request a major revision of the manuscript. I agree with rev. 1 that you should consider modifying your title, at least by eliminating "numerous". You should put special attention to the structure of your manuscript and a clear presentation of your findings. Please ensure also the correctness of your text (there are some spelling and style issues). Please note that both reviewers provided an annotated PDF which forms part of their review.

In particular, I would recommend to explain the statistic procedures you applied and to check/ improve the integration of your visual material (crosslinks and captions).

Reviewer 1 ·

Basic reporting

Acceptable. Specific comments were added on the pdf copy

Experimental design

Acceptable. Specific comments were added on the pdf copy

Validity of the findings

Discussion need to be expanded and conclusions revisited. See comments on the pdf copy and on General comments

Additional comments

Specific comments were annotated on the pdf copy.
General comments as follows.
259-261; 271-273. In my view, the most important finding of the paper is the association between genetic diversity and the maximum temperature in the warmest month (Fig. 5). However, such finding is not enough “exploited” (explained explicitly and stressed its importance) on the Discussion section. For example, is needed to state explicitly what is the trend: sites with warmer temperatures of the warmest month have populations with larger genetic biodiversity. Also, is needed to explain that such trend is counterintuitive regarding the North-South Axis: Northern populations have warmer temperatures of the Mmax. Explain why (northern locations are more arid than southern ones; notice the precipitation values on Table 7.)

261-263: Do not downplay your results. Yes, it might be obvious an association with soil and climate, but the specific pattern among populations with soil and climate is what matters here: how many authors have found an association as your Figure 5 for this rare and endangered species? Finally, I believe here the appropriate term is population (represented by the trees sampled), not provenance (which refer to the site)

273-276: “By contrast, the genetic diversity among the putative adaptive AFLPs was not significantly related to other variables. The relationships observed were probably not determined by selection, but by differences in the degree of isolation, which would influence gene flow and genetic drift”. Well, if climate does not matter (although that is what indicate Mmax), based on the lack of association to putative adaptive AFLPs variants, but what matters is distance among populations, then the discussion about this would need to be expanded. What if it is added some sort of spatial statistical analysis to explain the relationship between pairs of population distance and genetic diversity? Such spatial analysis might not be a good proof that what are you are guessing here (that the closer proximity of northern populations might facilitate gene flow among populations, and that increased the genetic diversity ) is a good possible explanation?

CONCLUSIONS
289-291 “First, at least three different ecotypes of Picea chihuahuana probably exist, as local adaptation may take place because of the different environmental conditions.” Well, yes, based on association with Mmax, but that is not supported by the lack of association with putative adaptive AFLPs variants.
It is needed to rethink this sentence as conclusion

301-303. “However, Picea chihuahuana grows in areas with special micro climate
conditions that are not easily modelled with simple macro climate models (Aguilar-Soto et al., 2015), but that can be recorded at local weather stations”
Is needed to clarify what you mean with “can be recorded at local weather stations”. Do you mean existing weather stations? The closest ones? That (likely) will be far away of the natural populations. In México, weather stations usually are either in urban areas or in flat agricultural areas; quite rarely in roughed mountains. Or do mean to install weather stations inside the natural P chihuahuana populations? Is needed to clarify that. In any case, this conclusion could be part of the discussion, rather than a conclusion.

Annotated reviews are not available for download in order to protect the identity of reviewers who chose to remain anonymous.

Reviewer 2 ·

Basic reporting

The manuscript reads clear in the beginning but becomes partly difficult to understand, actually not due to bad English but to unusal statements and unclear arguments.

The bibliography provided is well. However, it contains on the one hand general references that are not well linked and on the other hand lacks in-depth studies to clearly justify the intended approaches/methodologies.

Structure is well, figures and tables need to be improved (more attached). Only environmental data are shared, not the genetic ones.

Experimental design

The research is scope of PeerJ. Aims and goals are provided.
Methodological section lacks information or information is provided inconsitently (see attachement).

Validity of the findings

Soil data seem to be incomplete and biased due to a sampling strategy whose rationals are difficult to understand.
The general statistical approaches are acceptable but lack details.
Conclusions are rather part of discussion and do not perfectly draw implications out of the results.

Annotated reviews are not available for download in order to protect the identity of reviewers who chose to remain anonymous.

---

## Round 0.2 · Major Revisions

· Academic Editor

Major Revisions

Both reviewers agree that the manuscript substantially improved. However, especially reviewer 1 still expresses some doubts on the presentation of your work. Please make sure, to focus in your revised version on a consistent story and their statistical support.

Reviewer 1 ·

Basic reporting

Minor comments:

Line 270: "glm() (family = "binomial"), the step AIC() ". Sould not be something inside the paranthesis?

Line 343- "where sites with warmer Mtwm harbor populations larger genetic diversity. ". Hould be: "where sites with warmer Mtwm harbor populations with larger genetic diversity. ".

Line 374: "First, at least two different groups (natural proveniences) of P. chihuahuana probably exist ". I guess is better: "First, at least two different groups (clusters of natural populations) of P. chihuahuana probably exist ".

Experimental design

no comment

Validity of the findings

no comment

Additional comments

See my minor comments above

Reviewer 2 ·

Basic reporting

no comment

Experimental design

no comment

Validity of the findings

The findings are interesting and important for this very rare species.

However, after substantial improvement by the authors as compared to the previous version, I still feel very confused to understand the structure of the article, particularly when considering 74 that are seemingly not stringently condensated to the most important ones. I would guess here a table showing all obtained variables and highlighting those variables that have identified (after colinearity statistics, PCA,..) most important.

Second, most of the tables present the variables min, max, mean and SD values across populations. The scope of the paper, however, is to discriminate populations, which have been also found (2 clusters). Why do the authors not consistently present data of the two clusters, but rather average values, though there might be intersting variations in the subpopulations!

Third, it is stated that the number of trees measured varies between 17 and 53. As all variables and interpretation of the same do largely depend on the the sample size, I would like to see the number of N per population, apart of the already presented "TN".

Fourth, Fig. 5 looks great, but after deeper analysis, I doubt if this fact is worth to mention (s. last page of PDF)

Additional comments

Thank you for the papers studying adaptive AFLPs and the comment of BOTTLENECK!

I generally suggest a more strucutured approach by:
1. Decribe the variables obtained
2. decribing how you come to two clusters (genetic evidence)
3. give min, max, (or range), means SD only for the 2 clusters
4. perform the PCA or discriminant analysis to select most important variables
5. explain location wise differences based on the subset of variables and their implications for the populations.

All further suggestions, questions and comments are attached.

Annotated reviews are not available for download in order to protect the identity of reviewers who chose to remain anonymous.

---

## Round 0.3 · accepted · Accept

· Academic Editor

Accept

Both reviewers agree that the manuscript substantially improved, and I agree.

Reviewer 2 ·

Basic reporting

Dear Authors,

The current version looks much better now and thanks for the inclusion of the kriging model!
Structure and grammar is now fine, please find also a few changes in the PDF...rev3.

Experimental design

ok

Validity of the findings

ok

Additional comments

Nevertheless, the many data presented in tables can still be reduced to a more digestible format, but I would like to hand over this decision to the editor.

Apart of some correction in the text (s. PDF...rev3) two important facts/suggestions remain:

- The PCA-reduced set of variables was used to cluster populations, right? Consequently, why is the PCA plot then Figure 3 and not Figure 2?
Moreover, by simply astersiking or bolding the variables in the tables, the "Figure 3" is actually not necessary.

- Still, the presentation of Figure 6 is missleading in the current format. There should be at least axes breaks (both x and y) with a hint of significant Bonferroni correction!

Annotated reviews are not available for download in order to protect the identity of reviewers who chose to remain anonymous.